# A systematic review and meta-analysis on achievement emotions, working memory and student-teacher relationship during second language learning in primary school

Eugenio Trotta[1]*, Cristina Semeraro[2], Matteo Gatti[3], Gabrielle Coppola[2], Nicola Mammarella[3], Paola Palladino[1]

1 Applied Experimental Psychology Lab (AEP Lab), University of Foggia, Foggia, Italy, 2 Department of Education, Psychology, Communication, University of Bari Aldo Moro, Bari, Italy, 3 Department of Psychology, University of Chieti-Pescara, Chieti, Italy

* eugenio.trotta@unifg.it

## Abstract

Learning is a multidimensional process resulting from the interaction between cognitive and emotional factors within the learning context; in this respect the quality of the student-teacher relationship plays a significant role. Although the literature suggests that cognitive processes and emotions experienced during learning and task performing play a central role in academic achievement, it remains unclear how these factors interact with socio-affective factors in explaining academic performance, particularly in second language (L2) learning from primary school. This systematic review and meta-analysis focused on the studies that jointly or individually investigated the role of emotional factors (achievement emotions), socio-affective factors (student-teacher relationship) and cognitive factors (working memory) in L2 learning during primary school. Our sample contained 19 primary studies with 5,340 participants involved in at least one of the factors of our interest. 16 out of 19 studies were included in the meta-analysis. Our results showed a positive correlation between working memory and L2 learning, differentiated effects of achievement emotions, with a significant negative association with anxiety, and a small but positive association with enjoyment. The student-teacher relationship was supported only by qualitative evidence, however, showing a protective effect of emotional closeness to the teacher in the learning process in the presence of negative emotions such as anxiety. Findings support the importance of integrating cognitive, emotional, and relational factors to understand L2 learning in primary school. Further empirical research focusing on positive emotions and relational dynamics in different educational contexts is needed.

**Data availability statement:** All data underlying the findings are available within the paper, its Supporting Information files, and the Open Science Framework (OSF) repository (link provided in the manuscript).

**Funding:** This work was supported by the "Learning English as a second language in primary school: an investigation of metacognitive and socio-emotional factors on children's academic success and wellbeing - SELF_ENG", Research Project of Relevant National Interest 2022 (PRIN), part of the National Recovery and Resilience Plan (PNRR) also known as "Italia Domani", and component of the European Union's Next Generation EU recovery program (Protocol n. P2022TT8LH).

**Competing interests:** The authors have declared that no competing interests exist.

## Introduction

From early childhood onward, academic learning requires the integration of multiple cognitive, emotional, and relational factors [1,2]. Among cognitive factors, several studies have shown that working memory (WM) is a reliable predictor of academic performance, particularly in reading and mathematics [3,4]. At the same time, emotions profoundly influence the learning process, with effects that vary depending on the academic domain [5] and cultural context [6]. Emotions related to academic activities and outcomes are referred to as achievement emotions (AEs), defined as the emotions experienced during or in anticipation of learning-related task [7,8]. Another factor that significantly impacts learning is the quality of the student-teacher relationship (STR). Indeed, a relationship characterized by warmth and emotional support can act as a secure base that promotes students' autonomy and trust [9,10]. Although these three variables have been extensively investigated in relation to learning, their interaction and specific role in early educational contexts remain unclear, particularly in L2 learning, which offers opportunities for long-term cognitive, social, and academic development [11]. Therefore, this study aims to examine, through a systematic review and meta-analysis, how WM, AEs, and the quality of STR are associated with L2 learning in primary school children.

### Working memory and learning

Learning during childhood is a multifaceted process that engages a wide array of cognitive abilities [12], among which WM plays a crucial role. WM is a limited-capacity system responsible for the temporary maintenance and active manipulation of information required for complex cognitive tasks [13]. A robust body of research in educational psychology has documented a strong relationship between WM capacity and academic performance [3,14,15]. For instance, an efficient verbal WM facilitates language learning by allowing children to retain previously processed sentences while reading or listening [16,17]. Longitudinal evidence indicates that low WM scores in childhood predict both general learning difficulties and specific learning disorders, highlighting the strong predictive value of WM for later academic achievement [4,18]. WM also plays a fundamental role in language development and acquisition. Specifically, the phonological loop, which temporarily stores sounds and verbal information, is crucial for learning new words [19,20]. Verbal WM also supports the comprehension of syntactically complex sentences by enabling the integration of verbal elements as they are sequentially processed. Conversely, limitations in WM can impair linguistic development, as observed in children with specific language impairment (SLI), who typically exhibit reduced vocabulary and marked difficulties in both language comprehension and production [21,22]. Lastly, WM is closely related to domain-general cognitive functions such as sustained attention and self-regulation, both of which are recognized as strong predictors of academic achievement [23]. At the same time, these cognitive processes are closely intertwined with students' emotional and motivational experiences, which can either support or hinder the effective use of WM resources during learning activities [24]. Therefore, alongside cognitive

capacities like WM, emotional and motivational factors also play a significant role in shaping students' self-regulated learning processes [25].

## Achievement emotions and learning

Several studies highlighted that learning is strongly connected to students' emotions related to academic activities and outcomes, which can influence how students learn, their motivation, and their academic performance [8,26–30]. According to the *control-value theory* (CVT) [7,8], AEs depend on the perceived level of control students believe they have over an outcome (control) and the importance they assign to the task or goal (value). For example, students who believe they have strong mathematics skills (high control) and place great importance on academic success (high value) are more likely to experience positive emotions such as enthusiasm and pride (i.e., positive AEs). Conversely, when students perceive low control over an important task, they may experience anxiety or fear of failure (i.e., negative AEs). Most school-based studies on AEs have focused on mathematics. Raccanello et al. [31] found that Italian second-grade children experienced more positive emotions (fun) and fewer negative emotions (boredom and anxiety) than older fourth-grade students. They also reported more negative perceptions toward learning their native language (L1) than toward mathematics. In their longitudinal study, Lichtenfeld et al. [32] found that children's enjoyment progressively decreased over time, although this decline was not accompanied by a parallel increase in negative emotions. Pekrun et al. [33] found that positive emotions predicted adolescents' academic success in mathematics, while negative emotions were negatively associated with performance. These differences in study design (cross-sectional vs. longitudinal studies), age group (pupils vs. adolescents), and school subject (L1 vs. mathematics) are consistent with the domain-specific nature of AEs [5]. Furthermore, recent studies suggested that learning-related emotions can also be influenced by cultural context, as factors such as social expectations, educational practices, and values attributed to academic success differ across cultures and can shape students' emotional experience [6,34]. Camacho-Morles et al. [35] conducted a meta-analysis of 68 studies to investigate how AEs related to academic activities (rather than exclusively to outcomes) influenced students' academic performance. They found positive correlations between academic performance and enjoyment, and negative correlations with negative AEs such as anger, boredom, and frustration. Among the moderators of these effects were educational level (with stronger correlations in secondary school than in primary or university education) and, although less consistently, nationality and school subject. Therefore, AEs are influenced by individual characteristics (appraisals related to control and value experiences) as well as by the contextual factors operating at both proximal (e.g., task difficulty) and distal levels (e.g., educational systems). These factors may in turn be influenced by the quality of the student-teacher relationship [36].

## Student-teacher relationship and learning

Schools represent key social contexts for child development [37,38] and in recent decades, research has increasingly focused on the affective bond between teachers and students [39]. This line of research is rooted in attachment theory [40], which provides a theoretical framework for understanding the role of a caregiver's sensitivity to children's cues as a prerequisite for the development of secure relationships. Building on Robert Pianta's theoretical work [9], this framework has been extended to classroom relationships, shifting empirical attention from the purely institutional dimension to the affective quality of student-teacher interactions. Specifically, Pianta conceptualized STR as an attachment relationship. According to this perspective, such relationships depend on the previous affective experiences of both partners and therefore are influenced by the internal working models of attachment held by both the teacher and the student. Moreover, similarly to caregiver-child relationships, STR can function as a secure base from which students explore new learning opportunities, as well as a haven for regulating negative emotions within the school context. Since this conceptualization was introduced, several research supported the concept that a positive affective relationship with a teacher might promote learning and positive adaptation within the school context [41].

A robust body of research has shown that positive STRs are characterized by high levels of warmth and trust, and low negativity, which promote students' feelings of security [9,10,42]. Importantly, emotional security is considered essential for students' learning [42]. Pianta's model [9] proposes that this relationship is typically assessed across two dimensions, which can have a unique effect on students' behaviors and emotions: warmth and conflict [43]. Warmth refers to the level of involvement, closeness, affection, and openness of communication between a teacher and a student, while conflict refers to the extent to which the relationship is negative and problematic. High-quality STRs provide a supportive foundation for long-term student learning [44]. When students perceive that their teachers support them, they tend to perform better academically and experience greater school engagement [45]. Several studies found that students who have close relationships with their teachers are more likely to experience academic interest, engagement, achievement, self-efficacy, and motivation compared to students with more conflictual relationships [46–49]. In this regard, it is suggested that students who experience intimate relationships with teachers and form close bonds with the school may not only have increased opportunities to learn but may also utilize these opportunities more effectively, thanks to their motivation, self-regulation, and metacognitive ability during the learning process [50,51]. From this perspective, the quality of STR may also shape students' emotional experiences in the classroom and indirectly support the effective use of cognitive resources such as WM during learning activities [52]. This pattern is supported by meta-analytic evidence on primary school years [10,53] and by longitudinal designs over the primary school years [54].

## The present study

Evidence suggests that WM, AEs, and STR play a key role in learning; therefore, evaluating the impact of these factors on students' academic performance requires further investigation, particularly for L2 learning starting in primary school, given its wide-ranging developmental benefits. L2 learning at school represents a crucial factor for positive adaptation and long-term well-being across the life cycle, as it enables individuals to communicate effectively in an increasingly globalized world. Furthermore, L2 learning appears to be associated with improved cognitive and social skills, and consequently with better academic outcomes, not only in bilinguals but also in children raised monolingually who learn L2 at school [55,56]. This makes it particularly important, given its implications for future generations, to investigate which factors facilitate L2 learning in early educational context. Among these factors, the development of effective self-regulation in learning is a crucial goal. Sel-regulation refers to the ability to understand and manage one's learning processes to promote better outcomes, positive emotions and adaptive learning relationships. According to the Metacognitive and Affective Model of Self-Regulated Learning (MASRL) [57,58], the person-level characteristics, such as their cognitive and metacognitive abilities and their motivational and emotional states, interact in guiding the learning process. At the same time, the interaction between these individual factors and contextual characteristics (e.g., teachers and peers) influences not only the learning process itself but also learning outcomes [57–59]. On one hand, cognitive dimensions such as WM appeared to be involved in L2 across different ages and proficiency levels, both among Italian students learning English as L2 [60–62] and among students with other L1 learning various languages as L2 [63,64]. Another component of self-regulation is emotion regulation. This emotional competence might support students in coping with negative emotions, by, for example, reappraising a frustrating learning context towards a more adaptive meaning, or by prolonging positive emotion during the learning process [65]. Even though students' L2 learning appears to be modulated by emotions [66], learning-related emotions remain relatively understudied compared to the broader literature on L2 [33,67,68]. At the same time, emotions such as enjoyment, fear, anxiety, pride, and boredom, are raised not only as a function of tasks characteristics but also in relation to the quality of interactions with teachers and peers [8,69]. Nowadays meta-analytic evidence shows that positive STRs are related concurrently and longitudinally to school engagement and academic achievement, including L2, while the opposite pattern of results has been consistently reported for negative relationships [41,70].

Taken together, these findings suggest that L2 learning is shaped by the dynamic interplay between cognitive resources, emotional processes, and relational contexts, which jointly influence how learners engage with and benefit

from instructional experiences [71]. Considering the aforementioned literature, the present systematic review and meta-analysis aimed to investigate L2 learning in primary school children to better understand how L2 outcomes are associated with children's cognitive abilities, emotions, and the student-teacher relational context. Specifically, the aim of the current review was to describe and evaluate studies that examined how cognitive, emotional, and relational factors operate individually or in synergy in L2 learning. The aim of the meta-analysis was to quantify the effect sizes, thereby estimating the magnitude and direction of the associations between each of these variables and L2 learning. Potential moderators (e.g., educational setting, L1 transparency, or geographical location) were also considered to test whether they could explain any heterogeneity observed in the studies.

## Method

### Literature search and search strategy

The study was preregistered on the Open Science Framework (OSF: https://osf.io/9zwk5/overview?view_only=6560b446795b4e4db94eb4ae3396f74c – May 12th, 2025). The literature search was conducted in accordance with PRISMA guidelines [72]. First, we examined the preprint servers PsyArXiv, Open Science Framework, and PROSPERO, screening papers, preprints, and preregistered works on the same topics as ours. On November 22nd, 2024, we searched four different databases: Web of Science, PubMed, PsycArticles, and Scopus. We limited the identification to the topic section (title, abstract, keywords), using a combination of several terms related to AEs, WM, and STR on L2 learning in primary school (see the supporting information for details). Next, we searched further articles consulting the reference of the selected articles (backward analysis) and the studies that cited the same articles (forward analysis). The complete search strategy is provided in S1 File, while all effect size calculations and sensitivity analyses are reported in S2 File.

### Inclusion and exclusion criteria

We included any peer-reviewed article written in English or Italian, without any time limit in terms of publication, in line with our PEO: Population; Exposure; Outcome. Study participants (Population) had to be primary school students, typically aged between 5 and 11 years. The Exposure had to involve at least one of our variables of interest (AEs, WM, STR), either individually or in combination. The Outcome consisted of the association (positive, negative or neutral) between AEs, WM, and STR on L2 performance among primary school students. We excluded studies that did not satisfy those criteria. Furthermore, we excluded non-peer-reviewed publications, such as non-journal papers, editorials, dissertations, letters to authors, comments on published articles, and grey literature in general.

### Screening process

We used Rayyan to screen studies for eligibility [73]. After removing duplicates, two authors independently assessed the titles and abstracts of all identified articles against our criteria, with any disagreements resolved through subsequent discussion among all authors. The text of the remaining publications was screened by two authors. They performed reliability using the Cohen's Kappa (κ) to measure agreement between two reviewers. Cohen's Kappa was interpreted using Landis and Koch's convention [74] as Poor (< 0.00), Slight (0.00–0.20); Fair (0.21–0.40), Moderate (0.41–0.60), Substantial (0.61–0.80), and Almost Perfect (0.81–1.00) Agreement.

### Quality assessment

We evaluated the quality of the evidence and risk of bias for each study included in the systematic review, using methodology-specific assessment tools. Depending on the randomization of the sample, we used the Risk of Bias in Non-randomized Studies – of Interventions, Version 2 (ROBINS-I V2) [75] or the revised tool to assess risk of bias in randomized trials (RoB 2) [76]. The Newcastle-Ottawa Scale (NOS) [77] was used for observational studies. We computed the publication bias using the Trim and Fill method developed in Duval and Tweedie [78,79].

## Statistical analysis

We used MAJOR package in Jamovi version 2.6.44 and "metafor" package via R to perform a random-effects meta-analytic model [80–83]. The analysis was carried out using the Fisher r-to-z transformation correlation coefficient as the outcome measure. Sensitivity analysis was conducted using studentized residuals and Cook's distances to identify potential outliers or overly influential studies within the model [84]. We tested the heterogeneity of effect sizes at post-intervention using $I^2$ [85]. In addition, $tau^2$ and the Q-test for heterogeneity were reported [86]. Since several studies reported data on more than one time-points, we computed a combined effect across the time [87]. In addition to conducting separate meta-analysis for independent subgroups (WM, AEs, STR), we initially planned to examine associations between these variables. However, there was an inadequate number of comparisons to perform a meta-regression [87–89].

## Results

### Data extraction

We identified 322 potentially relevant papers: 98 from both from Web of Science and Scopus, 86 from PubMed, 40 from PsycArticles, and one article from a repository (OSF). See Fig 1 for the literature search process. After removing duplicates (n = 66), we screened the remaining articles from the topic section (title, abstract, keywords), excluding 175 out of 257 for several reasons: 133 for a different outcome (e.g., studies that use the acronym "L2" to indicate unrelated concepts, such as the French football second division, the second lumbar vertebra, or the second lumbar spinal nerve); 35

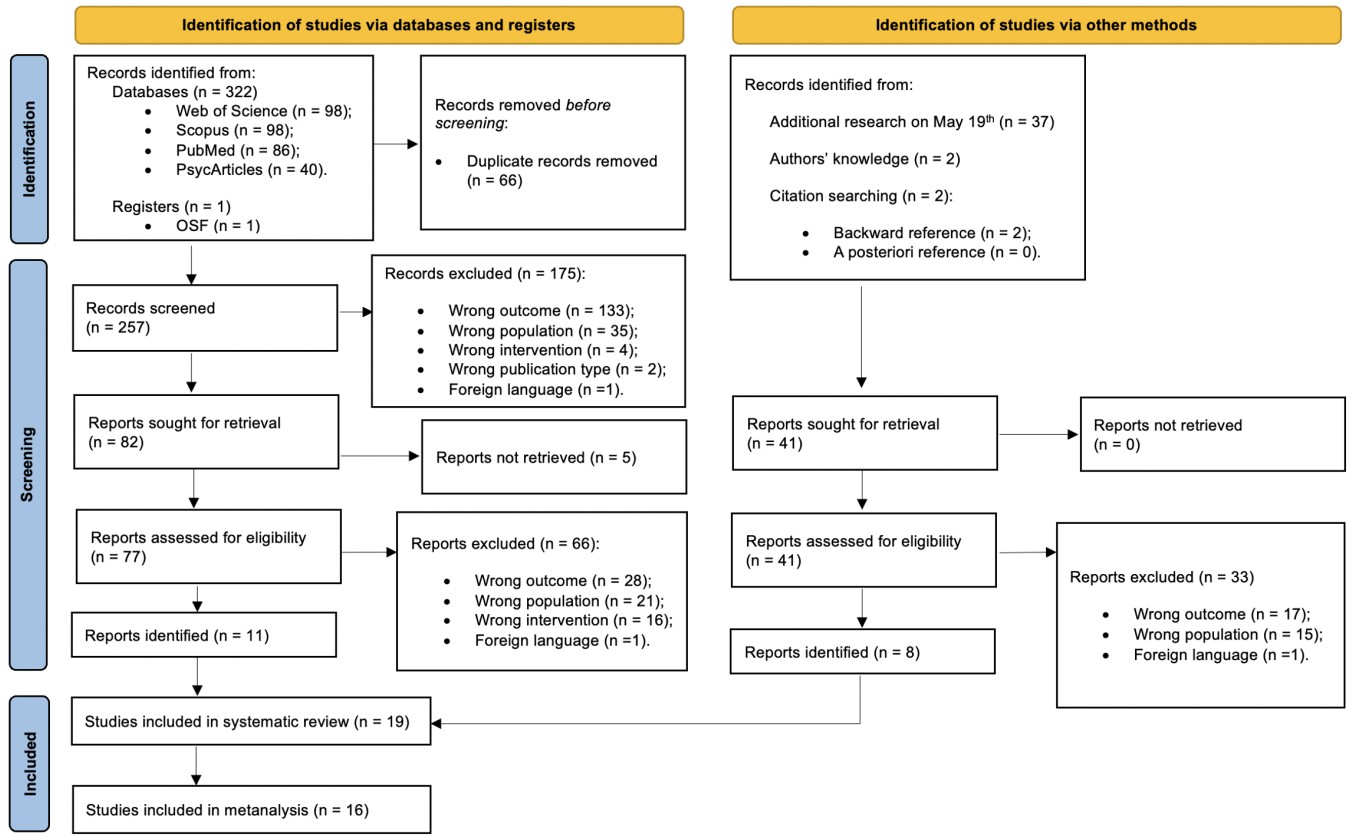

**Fig 1. PRISMA flowchart of the literature search and screening process.**

did not refer to a primary school sample; four studies had an intervention that was not in line with ours (e.g., interventions based on emotions, but not AEs). We also excluded two articles due to their publication type (a research protocol and an obituary) and a study written in Russian. Of the 82 remaining articles, we contacted the authors of seven unavailable articles via email or social media (ResearchGate) to request that their articles be shared. Five corresponding authors did not respond to the request. Hence, we reviewed the remaining 77 studies, according to the criteria defined in advance, leading to the exclusion of 66 papers. Several of those studies were excluded because their sample consisted of bilingual students, rather than L2 learners, or focused on one or more of our variables but on a different subject of study (e.g., mathematics). Regarding the final 11 papers, we investigated the presence of further articles of our interest by consulting their bibliography, through backward and forward searches. This allowed the identification of two articles. Two additional studies known to the authors were also included after verifying that they met the inclusion criteria. To ensure complete-ness, the database search was repeated on May 19th, 2025, identifying further four additional relevant studies. Regarding studies published in languages other than those included among the inclusion criteria (English and Italian), it is known that meta-analyses with language restrictions tend to overestimate treatment effects compared to those including stud-ies in all available languages, although they yield more precise results [90–92]. As illustrated in the PRISMA flowchart of the literature search and screening process (Fig 1), our search identified three scientific articles written in languages other than English and Italian, one of which was unavailable (requested from the authors but not received). We used a neural machine translation service (Google Translate) and a generative artificial intelligence chatbot (ChatGPT) to obtain literal and independent translations of the remaining two articles. The translated texts were subsequently reviewed by the research team to verify consistency with the original structure and key methodological information. Regardless of language restrictions, neither of these two studies addressed any of our variables of interest (WM, AEs, STR). Therefore, their exclusion from the data analysis was confirmed. We calculated a Cohen's Kappa (κ) of 0.84, almost perfect agree-ment. Finally, we selected 19 articles suitable for inclusion in systematic review, and 16 out of 19 in meta-analysis.

## Risk-of-bias assessment

Our systematic review included eight non-randomized or pseudo-randomized studies and 11 cross-sectional ones. There-fore, we assessed the quality of the evidence using the Risk of Bias in Nonrandomized Studies-of Interventions, Version 2 (ROBINS-I V2) for follow-up studies with a not or pseudo randomized experimental methodology [75], and the Newcastle-Ottawa Scale (NOS) for the cross-sectional studies [77]. With respect to cross-sectional studies, most of the included studies have a good or very good methodological quality, especially in terms of accuracy in exposure assess-ment and outcome assessment (see Table 1). The study of Li et al. [93] showed the lowest rating (2/10), with weaknesses across several domains, mainly due to limited methodological information and issues related to the reference sample. Since the risk of bias assessment does not necessarily imply the possibility that a study is biased but rather that it is susceptible to bias [94], we decided to preserve the unsatisfactory studies within our search. In regard to non-randomized or pseudo-randomized studies, no concern for bias in the result was highlighted, except for Chen [95] due to the lack of information regarding uncontrolled confounding. All other domains in Chen [95] and the other studies were classified as low risk of bias (see Fig 2).

## Description of the studies included

As reported in Table 2, the included studies had a sample size ranging from a minimum of 23 [96] to a maximum of 1,129 [97]. The study of Trotta et al. [98] represents a follow-up to Palladino et al. [99] on the role of certain variables on the same sample in the long and short time, respectively. Since Trotta et al. [98] and Palladino et al. [99] shared data from the same cohort, only Trotta et al. [98] was retained, as it provided the most complete and relevant dataset for the meta-analytic objectives, thereby avoiding data overlap and inflated effect size estimates. Therefore, the total sample comprised $N = 5,340$ ($M_{Age} = 9.58$; $SD = .59$; females $= 35.78\%$). We only considered the 23 children belonging to the regular group

**Table 1. Newcastle-Ottawa Scale adapted for Cross-Sectional studies (NOS-CS).**

| Cross-sectional studies (n = 12) | Selection | | | | Comparability | Outcome | | Total (0-10) * |
|---|---|---|---|---|---|---|---|---|
| | Representativeness of the sample | Sample size | Non-respondents | Ascertainment of the exposure | Adjustment of the outcome | Assessment of the outcome | Statistical test | |
| Dan et al. (2024) | 2 | 1 | 1 | 2 | 1 | 1 | 1 | 9 |
| Gillet et al. (2020) | 2 | 1 | 1 | 2 | 1 | 1 | 1 | 9 |
| Harrison and Boulet (2024) | 0 | 0 | 1 | 2 | 1 | 1 | 1 | 6 |
| Kersten (2022) | 1 | 1 | 1 | 2 | 1 | 1 | 1 | 8 |
| Li et al. (2019) | 0 | 0 | 0 | 1 | 0 | 1 | 0 | 2 |
| Liu and Hong (2021) | 2 | 1 | 1 | 2 | 1 | 1 | 1 | 9 |
| Matrić et al. (2019) | 2 | 1 | 1 | 2 | 1 | 1 | 1 | 9 |
| Trotta et al. (2024) | 2 | 1 | 1 | 2 | 1 | 1 | 1 | 9 |
| Tsang et al. (2025) | 1 | 1 | 1 | 1 | 0 | 1 | 1 | 6 |
| Wang et al. (2025) | 2 | 1 | 1 | 2 | 1 | 1 | 1 | 9 |
| Yang et al. (2018) | 0 | 0 | 1 | 2 | 0 | 1 | 1 | 5 |

Note. * 9–10 = Very Good; 7-8 = Good; 5-6 = Satisfactory; 0-4 = Unsatisfactory.

of Trebits et al. [96], as the immersive group (n = 16) was defined as bilingual (German and English). Furthermore, no demographic information about the sample of Li et al. [93] was reported, which mainly focused on the structure of digital tools in STR. We included all studies with children attending primary school as required by their country of origin, meaning that students from first grade to ninth grade were included. In terms of geographical distribution, one study was conducted in Canada [100], nine studies were in Europe [96–99,101–105] and as many in Asia [93,95,106–112]. Regarding L1, eight studies involved samples with an opaque native language: five with Chinese or Mandarin-speaking students [93,95,106–108,111,112], one Cantonese [110], and two French [100,102]. Three studies used moderately transparent languages, namely German [96,104] and Dutch [103]. Transparent languages were represented in six studies, with Italian [98,99,101], Norwegian [97], Slovak [105] and a multilingual group (Urdu, Hindi and Tagalog) mediated by English, all considered to be languages with relatively transparent orthography [109]. Most studies investigated English as an L2, with the exception of Gillet et al. [102] and Mingjia and Xian [109], which focused on Dutch and Chinese as an L2, respectively. 14 studies investigated the role of only one variable of our interest, namely: six focused on the role of WM [96,97,100,102,103,109]; six of the AEs [95,101,105,110–112] and two on the role of STR [93,107]. Among the remaining five included studies, three examined the role of WM and AEs jointly [98,99,106], while one included information about AEs and STR [108], and another one investigated both WM and STR [104]. Therefore, our systematic review included 10 studies on WM, 10 on AEs, and 4 on STR and their role in primary school students' L2 learning.

## Working memory and L2 Learning

Ten of the included studies investigated the role of WM in L2 learning [96–100,102–104,106,109], with a particular focus on verbal WM, assessed through the digit span backward task. Although only this type of WM was considered in the subsequent meta-analytic analyses, Goriot et al. [103] and Mingjia and Xian [109] also examined nonverbal and visual

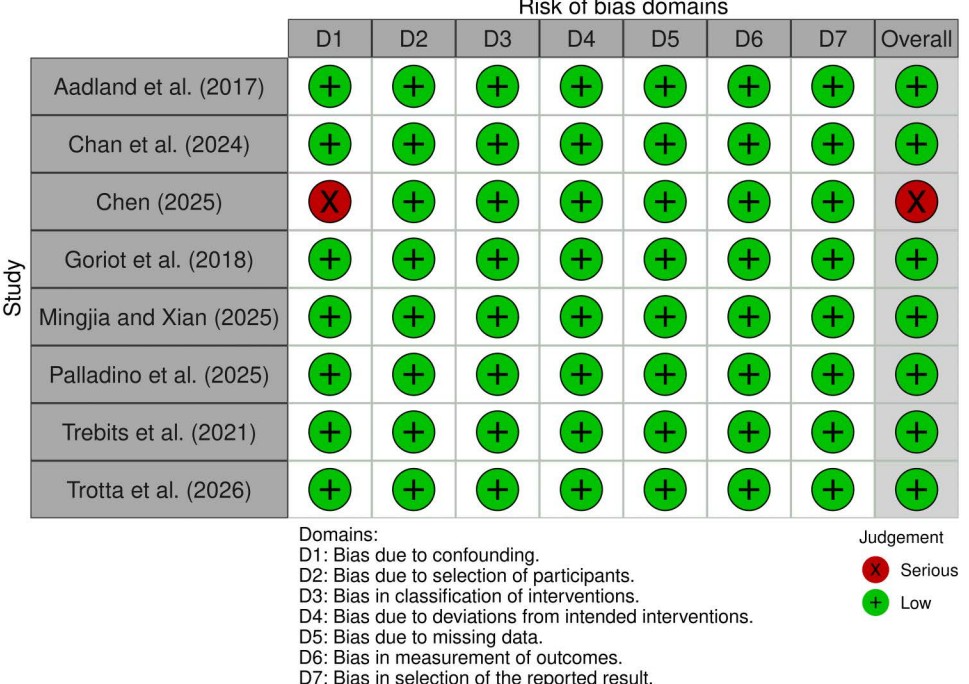

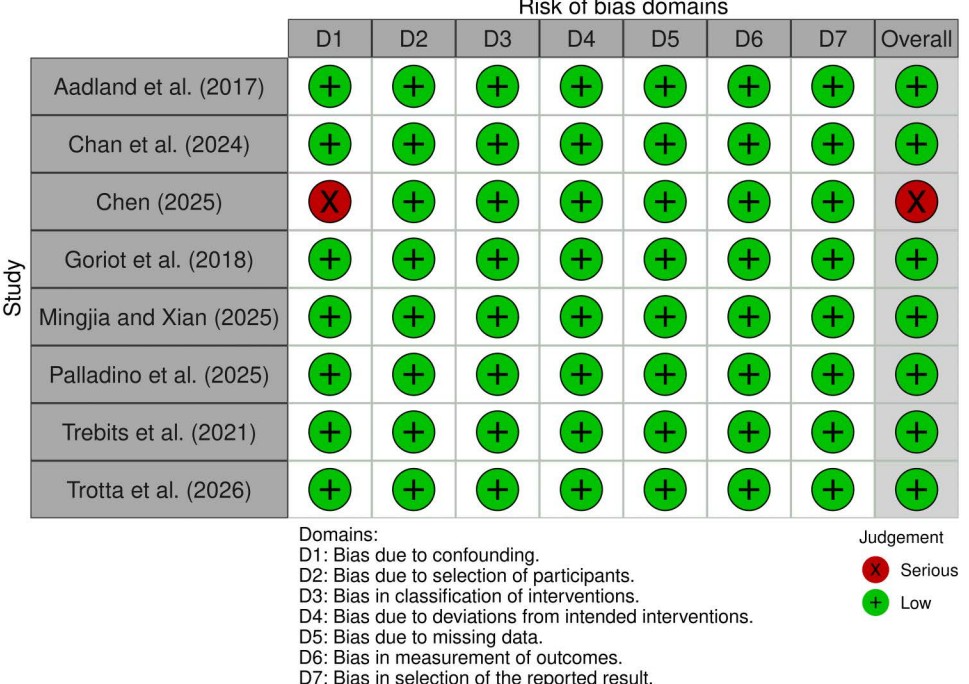

**Fig 2. Risk of Bias in Non-randomized Studies–of Interventions, Version 2 (ROBINS-I V2).**

WM, respectively. Most of the included studies examined the association between WM and L2 outcomes using Pearson's r correlations. In their two-stage longitudinal study, Palladino et al. [99] reported significant and consistent relationships over time between WM and academic performance in English vocabulary. Their findings were consistent with Goriot et al. [103] and Aadland et al. [97] who also identified gender differences. Specifically, females showed better performance in executive functions, including WM, whereas males demonstrated higher proficiency in English. No other studies reported significant gender differences nor identify significant relations between WM and L2 learning outcomes in cross-sectional [100] and longitudinal designs [106]. In contrast, several studies reported stronger correlations depending on factors such as familiarity with the L2 [98] or immersion time [96,102]. Mingjia and Xian [109] also examined the relationship between verbal WM and orthographic processing tasks in Chinese as an L2, reporting significant associations across all tasks. These findings support the role of WM in word recognition in an orthographically opaque language such as Chinese. Using structural equation modeling, Kersten [104] identified an indirect effect of WM on L2 grammar mediated by phonological awareness.

For the meta-analytic analysis, we computed a combined effect across time points for all studies reporting multiple measurements [87]. Two studies [96,99] were not included in the meta-analytic analyses but retained in the narrative synthesis to contribute to the broader systematic review. Although the study of Trebits et al. [96] met the inclusion criteria, it was excluded from the meta-analytic synthesis due to the absence of reported effect sizes. The available data did not allow extraction or estimation of the relationship between WM and L2 learning in the regular group. As previously reported, Trotta et al. [98] represents a follow-up to Palladino et al. [99], who investigated the role of several competences (including WM) in L2 learning using a short-term longitudinal design. Trotta et al. [98], on the other hand, examined the same constructs over a longer longitudinal period. Therefore, the two studies share two common data collection points. Avoiding an overlap that could lead to biased and unrealistic effect size estimates, only Trotta et al. [98] was included in the meta-analytic analyses. Gillet et al. [102] examined group differences in WM across multiple grade levels, reporting partial eta

**Table 2. Descriptive synthesis of the included studies.**

| Author(s) [alpha-betical order] | Aim | Sample | | | | | | | Investigated variable(s) [assessed with] | Results |
|---|---|---|---|---|---|---|---|---|---|---|
| | | N (female) | M(SD) | School grade | Country | L1 | L2 [assessed with] | | | |
| Aadland et al. (2017) | Investigate if executive functions (including WM) mediate the relation between physical activity and academic performance, which includes English as an L2 | 1.129 (541) | 10.2 (0.3) | 5th | Sogn og Fjordane (Norway) | Norwegian | English [Standardized Norwegian National test] | | WM [digit span backward] | Significant correlations were found between English as an L2 and WM, both at baseline and at follow-up |
| Chan et al. (2024) | Evaluate the impact of anxiety on L2 learning and the mediating effect of cognitive skills on it | 177 (67) | 94.31 months (8.31) | 2nd and 3rd | Hong Kong (China) | Chinese | English [English word reading] | | WM [Digit Span backward] AE (anxiety) [Child-rated foreign language reading anxiety, FLLA] | Pearson correlations highlighted a significantly negative relationship between English reading and foreign language anxiety, both T1 and T2. No significant relation between WM and L2 learning were found. |
| Chen (2025) | Evaluate the impact of anxiety on L2 learning using a digital-based educational setting | 30 (NA) | NA (NA) | NA | Taiwan | Mandarin Chinese | English [school-administrated English assessment] | | AE (anxiety) [Foreign Language Classroom Anxiety Scale, FLACS] | Splitting students into low and high anxiety level, results showed a L2 improvement for both groups, but more for student with high levels of anxiety. |
| Dan et al. (2024) | Evaluate how students' classroom relationships (including STR) contribute to L2 learning | 436 (208) | 10.43 (0.78) | 4th and 5th | China | Chinese | English [Self-report questionnaire] | | STR [Self-report questionnaire] | Correlational analyses showed that STR and L2 learning in both boys and girls were positively and significantly correlated with each other, more evident among boys. |
| Gillet et al. (2020) | Evaluate the role of executive functions (including WM) in L2 learning among students with Dutch immersion and traditional program | 391 (NA): 196 immersed 195 not immersed | 8,66 (0.28) | 1st, 2nd, 3rd, 6th | French-speaking part of Belgium | French | Dutch [adaptation of Expressive One-Word Picture Vocabulary test; adaptation of the Peabody Picture Vocabulary test, 3rd edition – PPVT-III-NL] | | WM [digit span backward] | For the first three years, no effects of group, time, and no interactions were found. However, after 6 years, children in the L2 immersion program performed significatively better than non-immersed children on WM tasks. |

*(Continued)*

| Author(s) [alphabetical order] | Aim | Sample | | | | | | Investigated variable(s) [assessed with] | Results |
|---|---|---|---|---|---|---|---|---|---|
| | | N (female) | M(SD) | School grade | Country | L1 | L2 [assessed with] | | |
| Goriot et al. (2018) | Evaluate the role of executive functions (including WM) in L2 learning among early English pupils and traditional L2 students | 204 (115) | 8.64 (0.39) | 1st, 5th, 8th | The Netherlands | Dutch | English [the Peabody Picture Vocabulary test, 4th edition – PPVT-4] | WM [Automated WM Assessment, AWMA; digit span backward] | Strong and significant correlations were found between L2 learning and WM, regardless of the type of education. Children in the older age groups scored significantly higher in L2 learning, and early English learners scored higher in English vocabulary. |
| Harrison and Boulet (2024) | Evaluate the role of executive functions (including WM) in L2 learning among students with French immersion program | 35 (21) | 8.65 (NA) | 3rd | Canada | French | English [the Peabody Picture Vocabulary test, 4th edition – PPVT-4) | WM [digit span backward] | No significant correlations were found between WM and English as an L2 |
| Kersten (2022) | Evaluate the relationships between variables on the meso-, micro-, and nano-level and L2 learning. | 93 (43) | 9.6 (NA) | NA | Germany | German | English [the British Picture Vocabulary Scale III, BPVS-III; Early Language and Intercultural Acquisition Studies II, ELIAS-II] | WM [digit span backward; letter-number sequencing] STR [Teacher Input Observation Scheme, TIOS] | No significant correlations were found between WM and L2 learning. The results showed that the L2 learning context, including teacher interactions and stimulation, influenced the students' L2 proficiency level. |
| Li et al. (2019) | Evaluate how the use of technology influences the way teachers and students interact with each other during English lessons | NA (NA) | NA (NA) | NA | China | Chinese | English [6 recordings of English as a foreign language lesson by teachers covering the same curricular unit, but with different levels of technology use] | STR [6 recordings of English as a foreign language lesson by teachers covering the same curricular unit, but with different levels of technology use] | The intensive use of technology did not facilitate L2 learning. The students' responses were strongly influenced by the teacher's teaching style. |

*(Continued)*

**Table 2.** (Continued)

| Author(s) [alpha-betical order] | Aim | Sample | | | | | | | Investigated variable(s) [assessed with] | Results |
|---|---|---|---|---|---|---|---|---|---|---|
| | | N (female) | M(SD) | School grade | Country | L1 | L2 [assessed with] | | | |
| Liu and Hong (2021) | Investigate the levels of students' enjoyment and anxiety in L2 learning in relation to gender and grade level | 267 (149) | 10.25 (0.6) | 4th and 5th | South China | Chinese | English [Questionnaire] | | AE (enjoyment; anxiety) [Self-report questionnaire] STR [Self-report questionnaire] | About a third of the participants felt anxious when speaking English, while more than half felt joyful in class, with no significant differences between grades and gender, except female students in 4th grade: more anxious than their male peers. Many students reported enjoying the class because they liked English and the teacher. |
| Matrić et al. (2019) | Investigate the role of anxiety on English as an L2 | 535 (277) | 11 to 14 years old (NA) | 6th – 9th | Slovenia | Slovak | English [final grades] | | AE [Foreign Language Classroom Anxiety Scale, FLACS] | Pearson correlations highlighted a significantly negative relationship between English final grades and foreign language anxiety |
| Mingjia and Xian (2025) | Evaluate the effects of executive functions (including WM) on L2 learning in Chinese as an L2 (CSL) students and native Chinese primary students. | 204 CSL 419 native students (females were 325 out of 623) | 9.93 (1.09) | 4th | Hong Kong (China) | Several (Urdu; Hindi; Tagalog). The communication between researcher and children was mediated with English | Chinese [Chinese vocabulary test, created ad hoc] | | WM [digit span backward] | Among CSL learners, significant correlations higher than those of native peers were found between WM and all the linguistic components investigated (recognition, reading, discrimination and orthographic choice of Chinese characters). |
| Palladino et al. (2025) | Evaluate the short-term role of metacognitive competence and AEs on L2 learning | 305 (48.5%) | 7.44 (0.59) | 2nd and 3rd | Calabria, Lombardia, Puglia (Italy) | Italian | English [English vocabulary test, created ad hoc] | | WM [Digit Span backward] AE (enjoyment; boredom; anxiety) [AEQ-ES] | Pearson correlations highlighted a slight relationship between WM and L2 learning, both at time 1 and at time 2. Regarding the AEs investigated, only boredom was found as a predictor of L2 learning |

*(Continued)*

**Table 2.** (Continued)

| Author(s) [alpha-betical order] | Aim | Sample | | | | | | | Investigated variable(s) [assessed with] | Results |
|---|---|---|---|---|---|---|---|---|---|---|
| | | N (female) | M(SD) | School grade | Country | L1 | L2 [assessed with] | | | |
| Trebits et al. (2021) | Investigate the cognitive differ-ence between monolingual and bilingual in L2 learning. We focused on monolingual. | 23 (14) | 9.59 (0.60) | 3rd | Germany | German | English [the British Picture Vocabulary Scale III, BPVS-III; Early Language and Intercultural Acquisition Studies II, ELIAS-II] | | WM [digit span back-ward; letter-number sequencing] | The bilingual children showed a marked improve-ment in WM in just one year, while the regular children did not. The difference between the groups was not present in 3rd, but it emerged clearly in 4th, in favor of the bilingual group. |
| Trotta et al. (2024) | Evaluate the role of AEs on L1 and L2 learning, at a cross-sectional level | 182 (46.82%) | 8.45 (0.55) | 2nd and 3rd | Lombardia, Puglia (Italy) | Italian | English [English vocabu-lary test, created ad hoc] | | AE (enjoyment; bore-dom; anxiety) [AEQ-ES] | Significant correla-tions were found, positive between L2 and enjoyment, and negative with bore-dom and anxiety. Regression analysis showed only anxiety significantly predicts performance in L2. |
| Trotta et al. (2026) | Evaluate the long-term role of metacogni-tive compe-tence and AEs on L2 learning | 305 (48.5%) | 7.44 (0.59) | 2nd and 3rd | Calabria, Lombardia, Puglia (Italy) | Italian | English [English vocabu-lary test, created ad hoc] | | WM [Digit Span backward] AE (enjoyment; bore-dom; anxiety) [AEQ-ES] | Significant correla-tions were found between WM and L2 learning, improving with age. Mixed effect model showed anxiety as a pre-dictor of L2 on the long-term. |
| Tsang et al. (2025) | Investigate students' perception of boredom in English homework | 95 (45) | 9.99 (0.57) | 5th | Hong Kong (China) | Cantonese | English [semi-structured interview] | | AE (boredom) [semi-structured interview] | 64% of the students considered English homework non boring |
| Wang et al. (2025) | Investigate the levels of students' enjoyment and anxiety in L2 learning | 381 (178) | NA | 4th and 6th | Xi'an (Shaanxi, China) | Chinese | English [Self-report questionnaire] | | AE (enjoyment; anxiety) [Self-report questionnaire] | Significant correla-tions were found, positive between L2 and enjoyment, and negative with anxiety. Latent profile analysis was computed, finding English results among students with the most enjoyment and the least anxiety were better than stu-dents with medium or high levels of anxiety and medium or low enjoyment. |

*(Continued)*

**Table 2.** (Continued)

| Author(s) [alphabetical order] | Aim | Sample | | | | | | | Investigated variable(s) [assessed with] | Results |
|---|---|---|---|---|---|---|---|---|---|---|
| | | N (female) | M(SD) | School grade | Country | L1 | L2 [assessed with] | | | |
| Yang et al. (2018) | Investigate the role of anxiety on L2 learning between high and low anxiety students, using game-based learning. | 43 (20) | 11-12 years (NA) | 4th | North Taiwan | Mandarin Chinese | English [English proficiency test, created ad hoc] | | AE (anxiety) [Foreign Language Classroom Anxiety Scale, FLACS] | Students with high anxiety levels had worse L2 learning than those with low anxiety levels |

*Note.* M = average age expressed in years; SD = standard deviation; L1 = native language; L2 = second language; AE = achievement emotion; STR = student-teacher relationship; WM = working memory; AEQ-ES = Achievement Emotion Questionnaire for Elementary School; NA = not available.

squared ($\eta p^2 = 0.10$). We converted it to Pearson's r ($r \approx .32$) using the square root transformation [113]. This conversion was performed only for Grade 6, as the effect was statistically significant ($p < .01$) and supported by Bayesian evidence in favor of the alternative hypothesis ($BF_{10} = 3.7$). In contrast, the analyses for Grades 1–3 yielded non-significant results ($p = .28$, $\eta p^2 < .01$, $BF_{01} = 5.0$), indicating substantial evidence for the null hypothesis. Since the inclusion of non-significant effects may bias meta-analytic estimates [87,114], these data were excluded from the quantitative synthesis, considering only grade 6. Therefore, a total of $k = 8$ studies were included in the meta-analysis. As shown in the forest plot (Fig 3), the observed Fisher r-to-z transformed correlation coefficients ranged from 0.09 to 0.78, with correlation coefficient based on the random-effects model equal to 0.29.

The true outcomes appeared to be heterogeneous ($Q(7) = 65.5060$, $p < 0.0001$, $tau^2 = 0.0462$, $I^2 = 90.96\%$). A sensitivity analysis was conducted. Studentized residuals and the Cook's distance indicated that one study [103] had a value larger than ± 2.73 and may represent a potential outlier or overly influential case. Neither the rank correlation nor the regression test indicated funnel plot asymmetry ($p = 0.9008$ and $p = 0.4891$, respectively). Excluding Goriot et al. [103], the average effect size decreased from $r = .279$ to $r = .218$, indicating a modest influence on the overall result (Fig 4). However, the main conclusion remained unchanged, supporting the robustness of the observed effect (see Sect 2 in the supporting information for diagnostic graphs of study influence).

These results were further supported by publication bias analyses, which showed no evidence of asymmetry. It is generally recommended to assess publication bias when at least ten studies are included in a meta-analysis [88]. However, it is also possible to detect publication bias with smaller studies (6–8 studies) using Tang's regression test [115,116]. This test revealed no evidence of publication bias ($t(6) = –0.14$, $p = .891$). The intercept estimate was $b = 0.32$ [–0.03, 0.68], indicating no substantial asymmetry in the funnel plot. For further confirmation, we compared the result with the funnel plot (Fig 5). Since the Trim and Fill method identified one trimmed study, we compared the overall effect size with the adjusted estimate. The corrected effect size ($r = .304$) was slightly higher than the observed effect size ($r = .279$), suggesting that results were not meaningfully influenced by publication bias.

## Achievement emotions and L2 Learning

Ten studies investigated the role of AEs in L2 learning. Four focused exclusively on the role of anxiety [95,105,106,112], one examined boredom alone [110], two considered both anxiety and enjoyment [108,111], and three assessed the role of enjoyment, boredom, and anxiety [98,99,101]. Chen [95] and Yang et al. [112], both involving Mandaring-speaking primary school students, used similar methodologies by dividing participants based on their level of anxiety related to L2 learning.

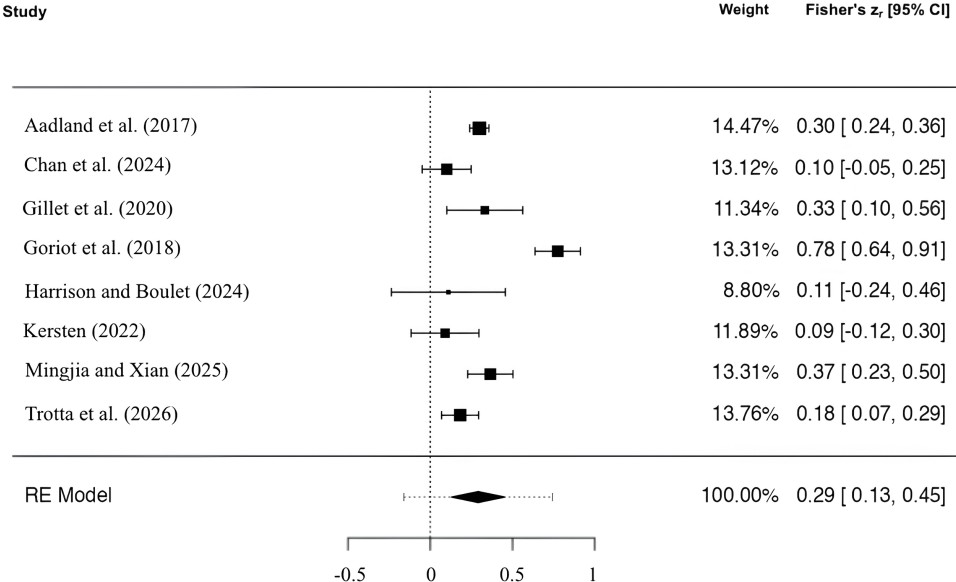

**Fig 3. Forest plot on WM and L2 Learning.**

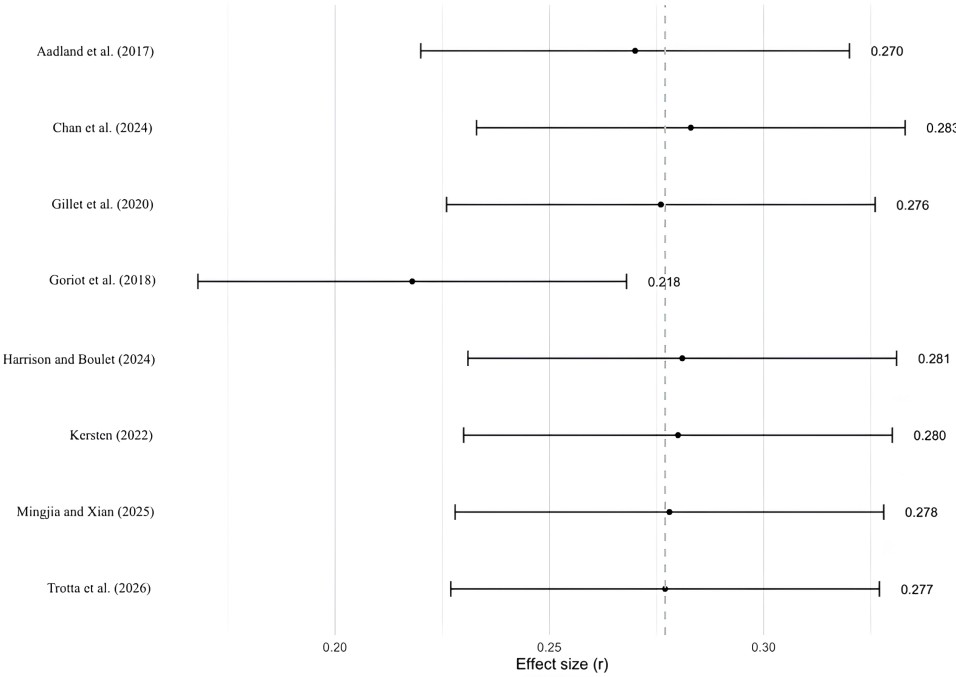

**Fig 4. Sensitivity Analysis on WM and L2 Learning: Leave-One-Out Metanalyses.**

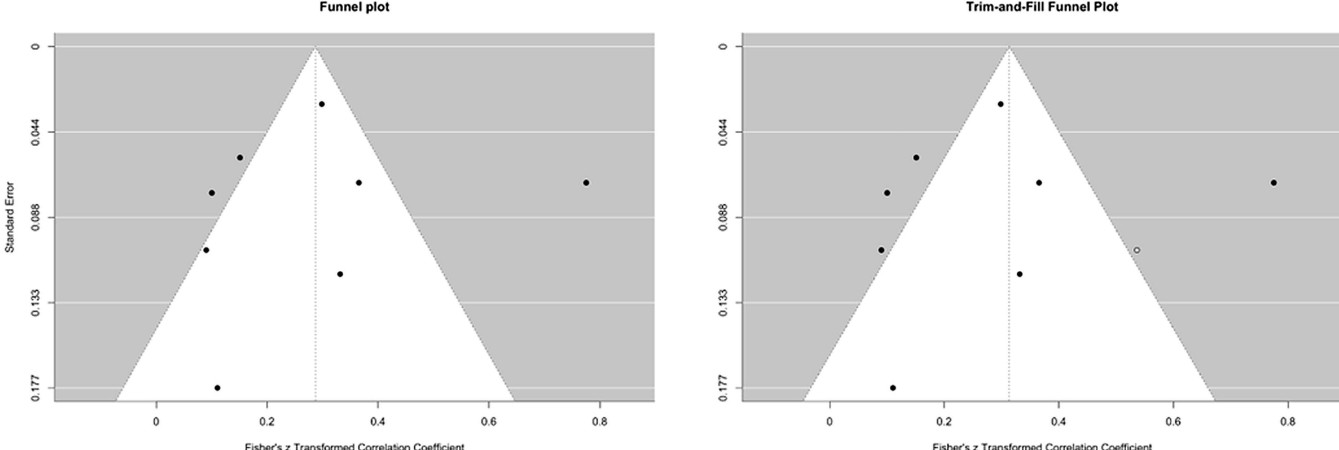

**Fig 5. Visual assessment of publication bias for WM and L2 Learning.**

Despite similar instructional approaches (e.g., game-based learning), their results were contradictory: Chen [95] reported better performance among students with high levels of anxiety, whereas Yang et al. [112] found poorer performance in the same group. Findings by Wang et al. [111] were consistent with Yang et al. [112]. Using a person-centered approach, they identified three classes of membership based on levels of enjoyment and anxiety in L2 learning, namely: "intense enjoyment and scant anxiety"; "medium enjoyment and medium anxiety"; "scant enjoyment and intense anxiety". Through a subsequent generalized linear model, Wang et al. [111] found that higher L2 performance was associated with a greater likelihood of being classified as having intense enjoyment and scant anxiety. In contrast, the "scant enjoyment and intense anxiety" group showed the highest levels of foreign language classroom anxiety (FLCA), followed in order by the "medium pleasure and medium anxiety" group and the "intense pleasure and low anxiety" group. Age-related differences also emerged, with older students (sixth grade) more likely to belong to the "scant enjoyment and intense anxiety" group compared to their fourth-grade peers. These results were consistent with Trotta et al. [101] and Liu and Hong [108]. This latter also considered the role of enjoyment and anxiety, finding 4th graders reported lower anxiety and higher enjoyment in L2 learning than 5th graders. Liu and Hong [108] also found a significant gender difference, with 4th grade female students reporting higher anxiety than males. Matrić et al. [105] also considered gender, specifically as a predictor of English as L2 anxiety in a hierarchical multiple regression model. However, gender was not found to be a statistically significant predictor. Regarding longitudinal studies, the results were not completely in line among them. In their two-stage study with Chinese students, Chan et al. [106] found that foreign language reading anxiety negatively predicted English reading performance at T1, and at T2 as an indirect effect of the English test and cognitive skills at T1. Palladino et al. [99] also evaluated the role of anxiety, with enjoyment and boredom in L2 learning in a short-term longitudinal design. The results of their multilevel model instead identified that only boredom was a strong significant predictor of English vocabulary at the early stage of L2 learning among Italian primary school students. However, the follow-up by Trotta et al. [98] identified anxiety as the only significant long-term predictor. Finally, Tsang et al. [110] was the only study to focus exclusively on boredom for English homework among Chinese 5th grade students. Their findings indicated that more than 60% of the students did not report boredom, suggesting that this positive result was mainly due to characteristics of the homework conveyed by the teacher and the relationship established with the student.

As for WM, we computed a combined effects for all studies with multiple comparison, outcome, and time point, and converted to Pearson's r where necessary (see Sect 2.1.2 in the supporting information for details). Palladino et al. [99] was excluded for the same reasons as in the previous section – sample overlap with Trotta et al. [98]. Due to its

descriptive design, Tsang et al. [110] was also not included in the meta-analytic analysis. The authors explored the perception of boredom in English tasks but did not provide data on the relationships between boredom and L2 learning outcome. Therefore, k = 8 studies on the role of AEs in L2 learning were included in the meta-analysis. All studies treated anxiety as an AE. Given the presence of studies that investigated more than one AE, we reported k = 14 studies, including four for enjoyment and two for boredom. As shown in Fig 6, we represented the result both stratified by emotions (Enjoyment, Boredom, Anxiety) and by valence (positive vs. negative).

Given the small number of studies available for each variable specific emotion (four for enjoyment and two for boredom), publication bias was assessed only for anxiety, for which k = 8 studies were available – a number generally considered sufficient for such analyses [116]. Tang's regression test [115] did not detect significant asymmetry (t(6) = 1.21, p = .27; b = –0.59, 95% CI: –1.11, –0.07). This result was further supported by the Trim and Fill method (Fig 7). The overall effect size of anxiety was r = –0.265 (z = –0.271). No trimmed results were highlighted, indicating no evidence of publication bias.

Regarding the sensitivity analysis, the true outcomes appeared to be heterogeneous (Q(7) = 89.20, p < 0.001, tau² = 0.0703, I² = 93.97%). As previously suggested by the risk of bias assessment (Fig 2), Chen [95] was identified as a potential outlier or overly influential study. Neither the rank correlation nor the regression test indicated funnel plot asymmetry (p = 0.7195 and p = 0.0704, respectively). Excluding Chen [95], the average negative effect size significantly increased. Matrić et al. [105] showed the most extreme effect size (r = −0.56), but its exclusion did not substantially alter the aggregate result. Likewise, none of the studies alone dramatically changed the result, so there were no outliers or influential cases (see Fig 8 and Sect 2.2.2 in the supporting information for diagnostic graphs of study influence).

## Student-teacher relationship and L2 Learning

Only four studies investigated the role of STR in L2 learning [93,104,107,108]. However, only two included quantitative data suitable for meta-analysis; therefore, the findings were assessed qualitatively. Dan et al. [107] reported gender-related differences. Specifically, their structural equation models showed that STR had both a direct effect and an indirect effect – mediated by metacognitive strategies – on English language proficiency, but only among boys. In contrast,

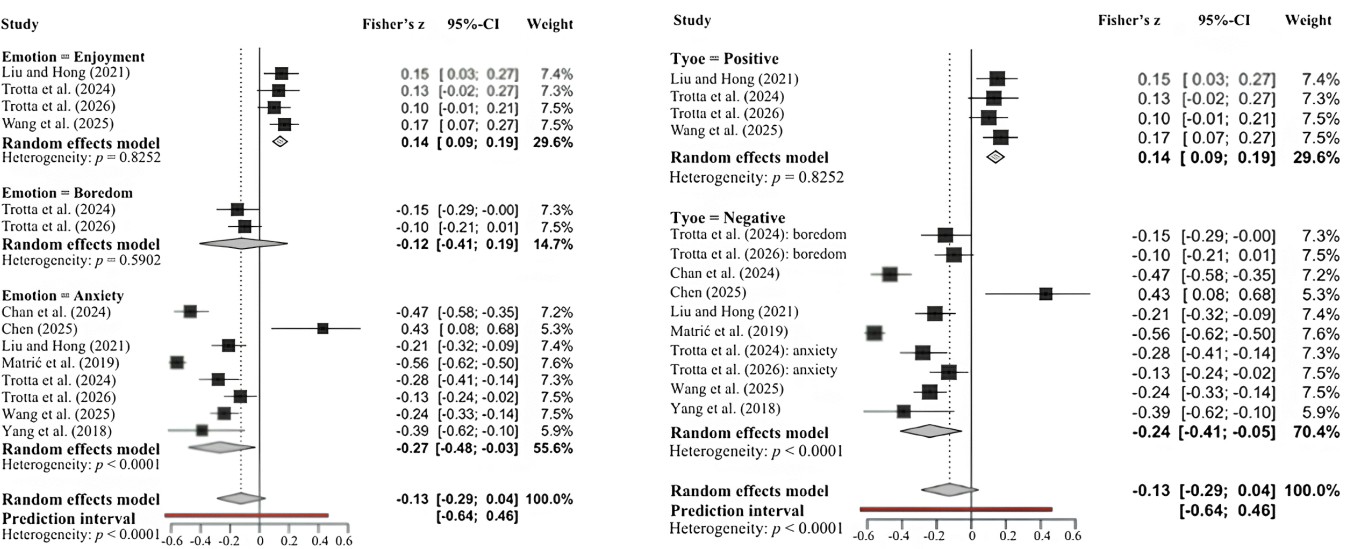

**Fig 6. Forest plot on AEs and L2 Learning: on the left, stratified metanalysis for emotions; on the right, stratified for valence.**

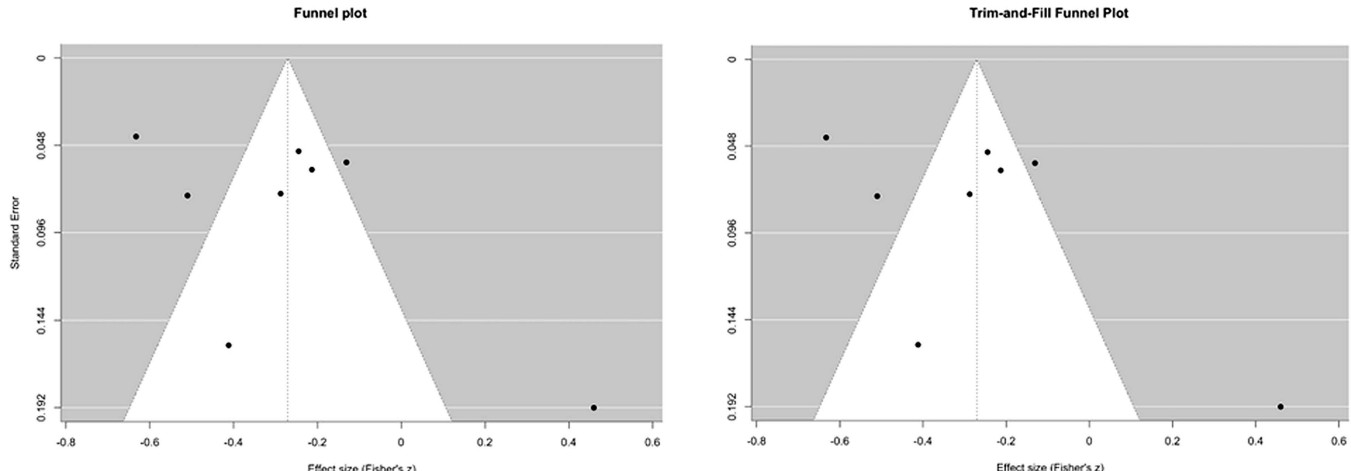

**Fig 7. Visual assessment of publication bias for anxiety and L2 Learning.**

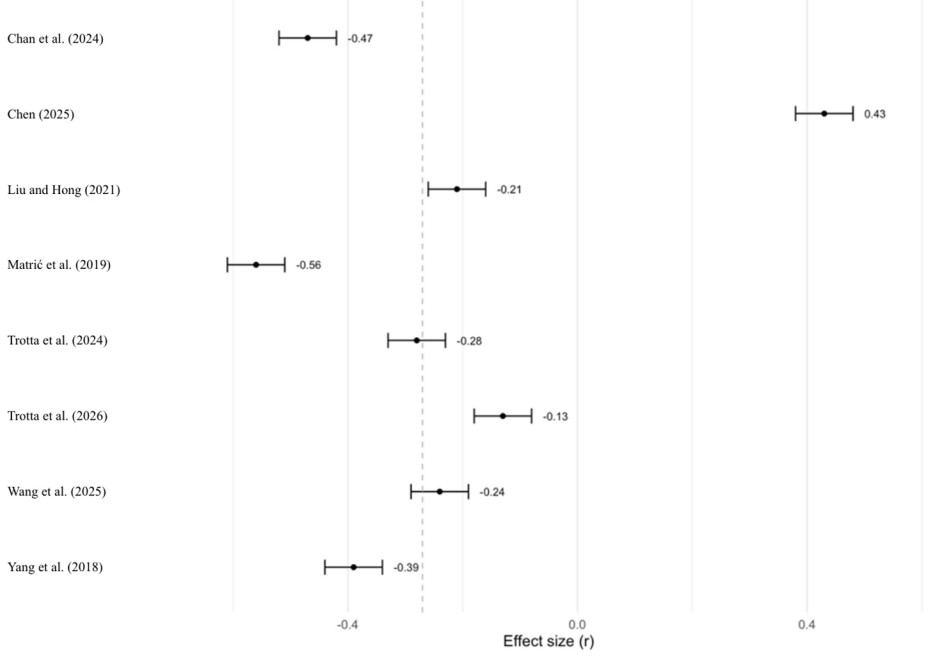

**Fig 8. Sensitivity Analysis on anxiety and L2 Learning: Leave-One-Out Metanalyses.**

Kersten [104], using a similar approach, did not find any gender-related differences. Although not explicitly focused on STR, Kersten's study highlighted that the quality and intensity of L2 input within teacher-student interactions were significantly associated with L2 grammar. The author therefore encouraged teachers to adopt "a particularly intensive use of L2 instructional techniques to make content comprehensible to the learners" [104]. Similarly, Li et al. [93] examined the impact of technology use on student-teacher interaction in English-as-L2 classrooms among Chinese primary school

students. By analyzing six video-recorded lessons – equally divided between high and low technology-using classrooms – the authors found that high technology use did not enhance interaction, and no significant effect in encouraging better language output from students was found. Finally, Liu and Hong [108], in their study on AEs, indirectly addressed the role of STR by examining the causes and effects of English language classroom anxiety and enjoyment. Among the 41 students who reported experiencing anxiety, 18 (43.9%) attributed it to teacher-related factors, such as fear of being questioned or punished, perceiving of the teacher as "terrifying", or fear or criticism. In contrast, among the 170 students who reported experiencing enjoyment, only 8 (4.7%) attributed this emotion to teacher-related factors.

## Discussion

The present systematic review and meta-analysis aimed to investigate the role of WM, AEs, and STR in L2 learning in primary school, with the goal of clarifying how these factors differentially contribute to L2 learning during this age. Our findings support the concept of learning as the result of a dynamic interaction between cognitive processes, emotional states, and contextual factors [57,58]. The analyses revealed a moderate positive effect of WM ($r = .29$) on L2 learning. This result is consistent with previous research highlighting the relevant role to WM, including verbal WM, in several academic skills, such as mathematics [14], linguistic processing, lexical acquisition and reading comprehension [60,63,117,118]. Although it was not possible to evaluate moderating or mediating effects, our meta-analysis also identified a significant negative effect of anxiety ($r = -.27$) and a positive effect of enjoyment ($r = .20$) on L2 learning, based on the limited number of available studies. Similarly, Cuder et al. [119] showed that the interaction between WM and mathematics anxiety influenced the mathematical students' performance, suggesting that anxiety could hinder the performance of students with high WM capacity. Regarding AEs perceived in L2 learning, the pattern observed in primary school students is consistent with the meta-analysis by Camacho-Morles et al. [35], which reported positive associations between academic performance and enjoyment, and negative associations with negative AEs (with exception of frustration). Using a person-centered approach, Radišić et al. [120] identified five emotional profiles in mathematics learning, showing that positive profiles (happy and moderate) were associated with better performance, whereas negative profiles (bored and anxious) were associated with poorer performance. Consistent with Frenzel et al. [6] and Loderer et al. [34], these findings also highlighted differences across cultural and educational contexts, supporting the domain-specificity of AEs [5]. According to control-value theory [7,8], such variability reflects the interplay between individual characteristics and contextual experiences. In this perspective, students' perceptions of the learning context may also be shaped by the affective quality of their relationship with the teacher. Although examined only qualitatively, STR appears to play a socio-affective role in L2 learning. Previous research has shown that positive STR are associated with better academic performance [41,70]. Our findings are consistent with this evidence, suggesting that perceived teacher support can either amplify or buffer performance-related anxiety. From an applied perspective, these findings underscore the importance of promoting positive and supportive classroom relationships from early schooling, with educational interventions that support the development of WM (e.g., metacognitive strategies, verbal span exercises) and reduce school anxiety regarding L2 learning. For example, teachers could reduce their students' anxiety and enhance their cognitive skills by actively engaging students in classroom activities, encouraging them to share their knowledge and recognizing themselves as contributors to knowledge. Such practices may reduce performance pressure and promote a psychologically safe learning environment in which students feel their competencies are recognized and valued [121]. We acknowledge several limitations of our work. First, although the number of included studies was adequate, particularly for WM and AEs (particularly anxiety), the number of studies examining STR was very limited (n = 4, of which only 2 with quantitative data), precluding meta-analytic synthesis for this variable. Therefore, conclusions regarding STR should be interpreted with caution and considered exploratory, as the limited evidence base reduces the strength and generalizability of these findings: further future research is needed. Second, heterogeneity was high across several analyses, likely due to differences in methodological characteristics, such as measurement tools, study duration, transparency or opacity of participants' native language, educational systems,

types of L2 learning (immersion vs. traditional), and grade levels. However, the number of studies for each variable of interest was insufficient to perform a meta-regression to assess potential combined effects or moderators (e.g., educational setting, L1 transparency, or geographical location). Third, although WM, AEs, and STR were analyzed separately, the initial aim was to examine their interaction. However, no studies were found that simultaneously investigated all three factors and their combined effects on L2 learning, despite theoretical support for their joint role [53,122,123]. Finally, most studies were conducted in Asian and European contexts, limiting the generalizability of the findings to other geographical areas. Therefore, future research with rigorous methodologies is needed to produce valid and replicable results that longitudinally investigate the interaction between all these factors, also considering socio-cultural and linguistic differences.

## Conclusions

To our knowledge, this is the first systematic review and meta-analysis to evaluate the role of WM, AEs, and STR in L2 learning in primary school. Rather than acting in isolation, these three variables appear to contribute to jointly shape L2 learning outcomes. The observed positive effect of WM suggests that students with greater cognitive resources are better able to cope with the processing demands of a new language. AEs appear to play a complementary role by influencing the conditions under which these cognitive resources are mobilized. Positive emotions, such as enjoyment, are associated with greater engagement, persistence, and openness to linguistic input, thereby facilitating learning processes. In contrast, anxiety may consume attentional resources, interfere with working memory functioning, and reduce learners' willingness to participate in communicative activities, ultimately hindering performance. Still, qualitative evidence suggests that students who perceive their teachers as supportive are more likely to feel confident and engaged, which may contribute to improvements in both affective states and cognitive functioning. As proposed by Vandenbroucke et al. [53], positive interactions with teachers can enhance students' trust, sense of security, and enjoyment, enabling them to persist and successfully complete even challenging tasks. Our findings could also inspire current multilingual education policies in the direction of emphasizing learner-centered, interactive, and inclusive approaches to language teaching. Regarding our topic, L2 learning depends not only on the acquisition of linguistic forms but also on the ability to use the language in different contexts and situations, manage communicative demands, and sustain engagement over time. Multilingual education policies should therefore adopt a holistic, multidimensional model of L2 learning in primary school, one that recognizes and fosters the reciprocal connections among cognition, emotion, and a supportive school environment [53,124]. References [93, 95–112] indicate studies included in the systematic review and meta-analysis.

## Supporting information

**S1 File. Additional information for Method section.**
(DOCX)

**S2 File. Additional information on quantitative analyses.**
(DOCX)

**S3 Table. PRISMA 2020 Checklist.**
(DOCX)

## Author contributions

**Conceptualization:** Eugenio Trotta, Gabrielle Coppola, Nicola Mammarella, Paola Palladino.

**Data curation:** Eugenio Trotta, Cristina Semeraro, Matteo Gatti.

**Formal analysis:** Eugenio Trotta, Cristina Semeraro, Matteo Gatti.

**Investigation:** Eugenio Trotta, Cristina Semeraro, Matteo Gatti.

**Methodology:** Eugenio Trotta, Cristina Semeraro, Matteo Gatti.

**Supervision:** Gabrielle Coppola, Nicola Mammarella, Paola Palladino.

**Writing – original draft:** Eugenio Trotta, Cristina Semeraro, Matteo Gatti, Gabrielle Coppola, Nicola Mammarella, Paola Palladino.

**Writing – review & editing:** Gabrielle Coppola, Nicola Mammarella, Paola Palladino.

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
