## [Decision Letter · Decision Letter 0]

24 Feb 2026

PONE-D-25-59942

A systematic review and metanalysis on achievement emotions, working memory and student-teacher relationship during second language learning in primary school.

PLOS One

Dear Dr. Trotta,

Thank you for submitting your manuscript to PLOS ONE. After careful consideration, we feel that it has merit but does not fully meet PLOS ONE’s publication criteria as it currently stands. Therefore, we invite you to submit a revised version of the manuscript that addresses the points raised during the review process.

We look forward to receiving your revised manuscript.

Kind regards,

Wei Lun Wong

Academic Editor

PLOS One

**Journal Requirements:**

Reviewers' comments:

Reviewer's Responses to Questions

**Comments to the Author**

1. Is the manuscript technically sound, and do the data support the conclusions?

Reviewer #1: Yes

Reviewer #2: Yes

2. Has the statistical analysis been performed appropriately and rigorously?

Reviewer #1: Yes

Reviewer #2: I Don't Know

3. Have the authors made all data underlying the findings in their manuscript fully available?

Reviewer #1: Yes

Reviewer #2: Yes

4. Is the manuscript presented in an intelligible fashion and written in standard English?

Reviewer #1: Yes

Reviewer #2: Yes

5. Review Comments to the Author

Reviewer #1:

Overall, the manuscript is structured with a clear and easy-to-follow flow, from the background, objectives, PRISMA methods, bias assessment, narrative synthesis, and meta-analysis, to the discussion and limitations. The English is generally up to academic standards, but still needs refinement to be more consistent and unambiguous, as PLOS ONE does not perform post-acceptance language editing.

Improvement Notes:

Unify terminology, particularly using a single, consistent form for "meta-analysis" and avoiding variations such as "metanalysis."

Correct several grammatical errors and unnatural phrasing, and break up long sentences for better readability.

Ensure abbreviations (WM, AEs, STR, L1, L2) are defined once at the outset and used consistently.

Clean up the formatting of lists and tables to avoid skipped numbering or layout artifacts.

Exercise caution in claims, especially for STR, as quantitative evidence is limited and relies heavily on qualitative synthesis.

Furthermore, the authors mention the use of AI to translate non-English or non-Italian articles. Translation procedures and verification should be explained concisely for transparency. Additionally, explain more explicitly how to detect and handle potential sample duplication in follow-up studies. Authors should also ensure a complete statement of data availability and supporting materials, including the search strategy and dataset extraction.

In conclusion, the manuscript is clear enough, but requires light to moderate editing for terminology consistency, minor grammar corrections, and formatting refinement before publication.

Reviewer #2: This systematic review and meta-analysis addresses a timely and important topic in educational psychology—the interplay between cognitive (working memory), emotional (achievement emotions), and socio-affective factors (student-teacher relationship) in second language learning during primary school. The manuscript makes a valuable contribution to the field by synthesizing evidence across 19 primary studies with over 5,300 participants.

Strengths:

The research addresses a significant gap by examining how multiple factors interact in L2 learning, moving beyond isolated variable approaches

Methodological rigor is evident in the systematic review process and meta-analytic techniques

Findings confirm expected associations (working memory positively correlates with L2 outcomes) while also revealing nuanced patterns (differential effects of anxiety versus enjoyment)

The qualitative synthesis of student-teacher relationship research provides valuable insights about the protective function of emotional closeness, particularly for anxious learners

The authors appropriately acknowledge limitations and suggest directions for future research

Minor Revisions Suggested:

Consider adding a brief discussion of implications for educational practice, particularly how teachers might leverage positive relationships to mitigate anxiety

A PRISMA flowchart would enhance transparency of the study selection process

The conclusion could be strengthened by explicitly connecting findings to the broader context of multilingual education policies

Overall, this well-conducted review advances our understanding of the multidimensional nature of L2 learning in primary school settings and deserves publication.

6. PLOS authors have the option to publish the peer review history of their article (what does this mean?). If published, this will include your full peer review and any attached files.

**Do you want your identity to be public for this peer review?** For information about this choice, including consent withdrawal, please see our Privacy Policy.

Reviewer #1: No

Reviewer #2: **Yes:** Sara Akram

td {border: 1px solid #cccccc;}br {mso-data-placement:same-cell;}

---

## [Author Response · Author response to Decision Letter 1]

17 Mar 2026

Point-by-point response letter PONE-D-25-59942: “A systematic review and metanalysis on achievement emotions, working memory and student-teacher relationship during second language learning in primary school”.

We thank the Academic Editor and the Reviewers for the time dedicated to our manuscript and for the constructive comments provided. Below, we present the Journal Requirements and the Reviewers’ comments to the Authors in black. The Authors’ responses are in blue. All changes in the revised manuscript are tracked with the “track changes” option.

Journal Requirements.

1. Please ensure that your manuscript meets PLOS ONE’s style requirements, including those for file naming. The PLOS ONE style templates can be found at https://journals.plos.org/plosone/s/file?id=wjVg/PLOSOne_formatting_sample_main_body.pdf and https://journals.plos.org/plosone/s/file?id=ba62/PLOSOne_formatting_sample_title_authors_affiliations.pdf

We confirm that our manuscript meets PLOS ONE’s style requirements. However, we cited Figures as “Figure” (e.g., Figure 1; Figure 2; etc.) instead of “Fig” as suggested by the manuscript body formatting guidelines. We modified it.

We apologize for the inconvenience. It was unintentional. As request, we are going to report as follow, when we resubmit:

This work was supported by the “Learning English as a second language in primary school: an investigation of metacognitive and socio-emotional factors on children’s academic success and wellbeing - SELF_ENG”, Research Project of Relevant National Interest 2022 (PRIN), part of the National Recovery and Resilience Plan (PNRR) also known as “Italia Domani”, and component of the European Union’s Next Generation EU recovery program (Protocol n. P2022TT8LH).

As request, we removed any funding-related text from the manuscript. Specifically, we removed the reference for the awards we received for our study (page 1, line 21) and the “Funding sources” section (page 23, line 599).

No reviewer comments included a recommendation to cite specific previously published works.

We confirm that our reference list is complete and correct. We did not identify cited papers that have been retracted.

Comments to the Authors from Reviewer 1.

Overall, the manuscript is structured with a clear and easy-to-follow flow, from the background, objectives, PRISMA methods, bias assessment, narrative synthesis, and meta-analysis, to the discussion and limitations. The English is generally up to academic standards but still needs refinement to be more consistent and unambiguous, as PLOS ONE does not perform post-acceptance language editing.

We thank Reviewer 1 for appreciating our work. As suggested, we refined the English of our manuscript to be more consistent and unambiguous with academic standards.

Improvement Notes:

• Unify terminology, particularly using a single, consistent form for “meta-analysis” and avoiding variations such as “metanalysis.”

We thank Reviewer 1 for the suggestion. We have consolidated the terminology with the term “meta-analysis.”

• Correct several grammatical errors and unnatural phrasing and break up long sentences for better readability.[ET1.1]

Thank you for the suggestion. We revised the manuscript to improve grammatical accuracy, reduce long sentences, and enhance readability.

• Ensure abbreviations (WM, AEs, STR, L1, L2) are defined once at the outset and used consistently.

We ensure that all abbreviations are defined upon first use and used consistently.

• Clean up the formatting of lists and tables to avoid skipped numbering or layout artifacts.[ET2.1]

Thank you for the suggestion. We revised the manuscript to ensure consistent layout, correct numbering, and the removal of any spacing or formatting artifacts.

• Exercise caution in claims, especially for STR, as quantitative evidence is limited and relies heavily on qualitative synthesis.

We thank Reviewer 1 for the observation. Indeed, the evidence regarding STR is limited to qualitative data. We revised the manuscript, adopting more cautious language in interpreting these results.

• Furthermore, the authors mention the use of AI to translate non-English or non-Italian articles. Translation procedures and verification should be explained concisely for transparency.

We thank Reviewer 1 for the comment. As suggested, we clarified in the manuscript the translation and review procedures for articles not available in English or Italian. Briefly, translations were obtained using two independent tools, and the resulting texts were subsequently checked by the research team for consistency with the original structure and key methodological information.

• Additionally, explain more explicitly how to detect and handle potential sample duplication in follow-up studies. Authors should also ensure a complete statement of data availability and supporting materials, including the search strategy and dataset extraction.

As suggested, we detailed how we handled papers with the same sample in "Description of the studies included" section. All data analysis and supporting materials, including the search strategy and dataset extraction, were previously shared within the Supporting Information. We forgot to mention it in the main manuscript. We thank the Reviewer 1 for the suggestion, giving us the opportunity to report it (see "Literature Search and search strategy" section).

In conclusion, the manuscript is clear enough, but requires light to moderate editing for terminology consistency, minor grammar corrections, and formatting refinement before publication.

We thank the Reviewer 1 for the time dedicated to our manuscript and for the constructive comments provided.

Comments to the Authors from Reviewer 2.

This systematic review and meta-analysis addresses a timely and important topic in educational psychology—the interplay between cognitive (working memory), emotional (achievement emotions), and socio-affective factors (student-teacher relationship) in second language learning during primary school. The manuscript makes a valuable contribution to the field by synthesizing evidence across 19 primary studies with over 5,300 participants.

Strengths:

• The research addresses a significant gap by examining how multiple factors interact in L2 learning, moving beyond isolated variable approaches

• Methodological rigor is evident in the systematic review process and meta-analytic techniques

• Findings confirm expected associations (working memory positively correlates with L2 outcomes) while also revealing nuanced patterns (differential effects of anxiety versus enjoyment)

• The qualitative synthesis of student-teacher relationship research provides valuable insights about the protective function of emotional closeness, particularly for anxious learners

• The authors appropriately acknowledge limitations and suggest directions for future research

We thank Reviewer 2 for appreciating our work.

Minor Revisions Suggested:

• Consider adding a brief discussion of implications for educational practice, particularly how teachers might leverage positive relationships to mitigate anxiety

We thank Reviewer 2 for his suggestion. We expanded the Discussion section by adding a practical example of how teachers can help reduce anxiety and support their students’ cognitive competences by promoting an emotionally safe and engaging environment.

• A PRISMA flowchart would enhance transparency of the study selection process

We agree with Reviewer 2. We previously reported our PRISMA flowchart of the literature search and screening process as Fig 1. Since the manuscript body formatting guidelines state “figures should be uploaded separately as individual files”, you can consult our PRISMA flowchart inside the file named “Figures”.

• The conclusion could be strengthened by explicitly connecting findings to the broader context of multilingual education policies.

We thank Reviewer 2 for the suggestion. As suggested, we integrated our findings into multilingual education policies.

Overall, this well-conducted review advances our understanding of the multidimensional nature of L2 learning in primary school settings and deserves publication.

We thank the Reviewer 2 for the time dedicated to our manuscript and for the constructive comments provided.

---

## [Decision Letter · Decision Letter 1]

1 Apr 2026

PONE-D-25-59942R1A systematic review and meta-analysis on achievement emotions, working memory and student-teacher relationship during second language learning in primary school.PLOS One

Dear Dr. Trotta, 

Thank you for submitting your manuscript to PLOS ONE. After careful consideration, we feel that it has merit but does not fully meet PLOS ONE’s publication criteria as it currently stands. Therefore, we invite you to submit a revised version of the manuscript that addresses the points raised during the review process.

We look forward to receiving your revised manuscript.

Kind regards,

Wei Lun Wong

Academic Editor

PLOS One

Journal Requirements:

Reviewers' comments:

Reviewer's Responses to Questions

**Comments to the Author**

1. If the authors have adequately addressed your comments raised in a previous round of review and you feel that this manuscript is now acceptable for publication, you may indicate that here to bypass the “Comments to the Author” section, enter your conflict of interest statement in the “Confidential to Editor” section, and submit your "Accept" recommendation.

Reviewer #1: All comments have been addressed

Reviewer #2: All comments have been addressed

2. Is the manuscript technically sound, and do the data support the conclusions?

Reviewer #1: Partly

Reviewer #2: Yes

3. Has the statistical analysis been performed appropriately and rigorously? 

Reviewer #1: Yes

Reviewer #2: Yes

4. Have the authors made all data underlying the findings in their manuscript fully available?

Reviewer #1: No

Reviewer #2: Yes

5. Is the manuscript presented in an intelligible fashion and written in standard English?

Reviewer #1: No

Reviewer #2: Yes

6. Review Comments to the Author

Reviewer #1: This manuscript presents a systematic review and meta-analysis examining the relationship between working memory, achievement emotions, and student–teacher relationships in second language (L2) learning among primary school students. The topic is relevant and timely, as understanding the interaction between cognitive, emotional, and contextual factors is important for improving language learning outcomes. Overall, the manuscript demonstrates potential to contribute to the literature in educational psychology and second language acquisition.

The study applies an appropriate systematic review methodology and follows PRISMA guidelines for study selection. The use of meta-analytic techniques, including random-effects models, heterogeneity analysis, and publication bias assessment, indicates an effort to ensure methodological rigor. The integration of three important constructs (working memory, achievement emotions, and student–teacher relationships) provides a multidimensional perspective that is valuable for both researchers and practitioners.

However, several issues should be addressed to improve the scientific quality and clarity of the manuscript.

First, although the statistical analyses are generally appropriate, the number of studies examining student–teacher relationships appears limited, which may affect the strength and generalizability of the conclusions related to this variable. The authors are encouraged to discuss this limitation more explicitly and clarify how it may influence interpretation of the results.

Second, the manuscript would benefit from clearer reporting regarding data availability. According to PLOS data policy, all data underlying the findings should be fully accessible either within the manuscript, supporting information files, or a public repository. The authors should provide explicit information about where the extracted dataset, effect size calculations, and analysis files can be accessed.

Third, the manuscript requires careful language editing to improve clarity and readability. Several grammatical issues, repeated phrases, and awkward sentence structures are present throughout the text. Professional proofreading is recommended to ensure the manuscript meets the standard of academic English expected for publication.

Additionally, the authors may consider improving the consistency of terminology and providing clearer explanations of the theoretical framework linking cognitive and emotional factors in L2 learning. Clarifying the rationale for inclusion criteria and providing more detailed justification for methodological choices would further strengthen the manuscript.

Despite these concerns, the study addresses an important research topic and applies appropriate methodological procedures. With revisions focusing on language clarity, transparency of data availability, and clearer discussion of limitations, the manuscript has the potential to make a meaningful contribution to the field.

No concerns related to research ethics, plagiarism, or duplicate publication were identified.

Overall recommendation: minor revision.

Reviewer #2: This is an excellent study. The methods are rigorous, the analyses are appropriate, and the conclusions are well supported. I have no further suggestions—congratulations on this valuable contribution.

7. PLOS authors have the option to publish the peer review history of their article (what does this mean?). If published, this will include your full peer review and any attached files.

Reviewer #1: No

Reviewer #2: **Yes:** sara Akram

---

## [Author Response · Author response to Decision Letter 2]

20 Apr 2026

Point-by-point response letter PONE-D-25-59942R1: “A systematic review and meta-analysis on achievement emotions, working memory and student-teacher relationship during second language learning in primary school”.

We thank the Academic Editor and the Reviewers for the time dedicated to our manuscript and for the constructive comments provided. Below, we present the Reviewers’ comments to the Authors in black. The Authors’ responses are in blue. All changes in the revised manuscript are tracked with the “track changes” option.

Comments to the Authors from Reviewer 1.

This manuscript presents a systematic review and meta-analysis examining the relationship between working memory, achievement emotions, and student–teacher relationships in second language (L2) learning among primary school students. The topic is relevant and timely, as understanding the interaction between cognitive, emotional, and contextual factors is important for improving language learning outcomes. Overall, the manuscript demonstrates potential to contribute to the literature in educational psychology and second language acquisition.

The study applies an appropriate systematic review methodology and follows PRISMA guidelines for study selection. The use of meta-analytic techniques, including random-effects models, heterogeneity analysis, and publication bias assessment, indicates an effort to ensure methodological rigor. The integration of three important constructs (working memory, achievement emotions, and student–teacher relationships) provides a multidimensional perspective that is valuable for both researchers and practitioners.

However, several issues should be addressed to improve the scientific quality and clarity of the manuscript.

We thank Reviewer 1 for the time dedicated to our manuscript and for appreciating our work.

First, although the statistical analyses are generally appropriate, the number of studies examining student–teacher relationships appears limited, which may affect the strength and generalizability of the conclusions related to this variable. The authors are encouraged to discuss this limitation more explicitly and clarify how it may influence interpretation of the results.

We thank Reviewer 1 for the comment. We agree that the limited number of studies investigating the student–teacher relationship (STR) represents a constraint for the strength and generalizability of the conclusions. As reported in the first limit (page 29, row 548), only four studies addressed STR, and only two provided quantitative data suitable for meta-analysis. For this reason, we treated the evidence on STR exclusively at a qualitative level and explicitly framed these findings as exploratory. To address this concern, we strengthened the discussion of this limitation in the manuscript by clarifying that:

● the small number of studies prevents robust generalization of the STR findings;

● conclusions regarding STR should be interpreted with caution;

● further research is needed to provide more solid and generalizable evidence on this variable.

Second, the manuscript would benefit from clearer reporting regarding data availability. According to PLOS data policy, all data underlying the findings should be fully accessible either within the manuscript, supporting information files, or a public repository. The authors should provide explicit information about where the extracted dataset, effect size calculations, and analysis files can be accessed.

We totally agree with Reviewer 1 on the importance of transparency and data availability. As already indicated in the manuscript (see “Literature search and search strategy”, line 195), all data underlying the findings are fully accessible in the Supporting Information files. Specifically:

● The complete search strategy (including research question development, use of synonyms, and Boolean operators across all databases) is provided in S1 File “Additional information for Method section”;

● All effect size calculations, as well as sensitivity analyses, are reported in S2 File “Additional information on quantitative analyses”.

Furthermore, in line with PLOS data-sharing policies, the study was preregistered on a public repository (Open Science Framework). For the purposes of double-blind peer review, an anonymous access link is currently provided in the manuscript; this will be replaced with a fully accessible public link upon publication.

Given the extent and technical detail of the search strategy and quantitative analyses, these materials are not included in full within the main manuscript but are entirely available in the supplementary files to ensure transparency and reproducibility. To further improve clarity, we strengthened the description of data availability in the manuscript.

Third, the manuscript requires careful language editing to improve clarity and readability. Several grammatical issues, repeated phrases, and awkward sentence structures are present throughout the text. Professional proofreading is recommended to ensure the manuscript meets the standard of academic English expected for publication.

We thank Reviewer 1 for the recommendation. We declare that we used ChatGPT (OpenAI, San Francisco, CA, USA) to improve the English language and the writing quality. After using it, we reviewed and edited the content as needed and took full responsibility for the content of the publication.

Additionally, the authors may consider improving the consistency of terminology and providing clearer explanations of the theoretical framework linking cognitive and emotional factors in L2 learning. Clarifying the rationale for inclusion criteria and providing more detailed justification for methodological choices would further strengthen the manuscript.

We improved clarity, refined sentence structure, ensured terminological consistency, and strengthened the theoretical framework by explicitly highlighting the interaction between cognitive, emotional, and relational factors throughout the Introduction.

Despite these concerns, the study addresses an important research topic and applies appropriate methodological procedures. With revisions focusing on language clarity, transparency of data availability, and clearer discussion of limitations, the manuscript has the potential to make a meaningful contribution to the field.

No concerns related to research ethics, plagiarism, or duplicate publication were identified.

Overall recommendation: minor revision.

We thank Reviewer 1 for the time dedicated to our manuscript and for appreciating our work.

Comments to the Authors from Reviewer 2.

This is an excellent study. The methods are rigorous, the analyses are appropriate, and the conclusions are well supported. I have no further suggestions—congratulations on this valuable contribution.

We thank Reviewer 2 for the time dedicated to our manuscript and for appreciating our work.

---

## [Decision Letter · Decision Letter 2]

11 May 2026

A systematic review and meta-analysis on achievement emotions, working memory and student-teacher relationship during second language learning in primary school.

PONE-D-25-59942R2

Dear Dr. Trotta,

We’re pleased to inform you that your manuscript has been judged scientifically suitable for publication and will be formally accepted for publication once it meets all outstanding technical requirements.

Kind regards,

Muhammad Zammad Aslam, Ph.D.

Academic Editor

PLOS One

Additional Editor Comments (optional):

Reviewers' comments:

Reviewer's Responses to Questions

**Comments to the Author**

1. If the authors have adequately addressed your comments raised in a previous round of review and you feel that this manuscript is now acceptable for publication, you may indicate that here to bypass the “Comments to the Author” section, enter your conflict of interest statement in the “Confidential to Editor” section, and submit your "Accept" recommendation.

Reviewer #2: All comments have been addressed

2. Is the manuscript technically sound, and do the data support the conclusions?

Reviewer #2: Yes

3. Has the statistical analysis been performed appropriately and rigorously? 

Reviewer #2: I Don't Know

4. Have the authors made all data underlying the findings in their manuscript fully available?

Reviewer #2: Yes

5. Is the manuscript presented in an intelligible fashion and written in standard English?

Reviewer #2: Yes

6. Review Comments to the Author

Reviewer #2: The authors have thoroughly addressed all previous concerns. The manuscript is significantly improved in clarity, methodology, and reporting. I am pleased to recommend acceptance for publication. Good luck with the next steps.

7. PLOS authors have the option to publish the peer review history of their article (what does this mean?). If published, this will include your full peer review and any attached files.

Reviewer #2: **Yes:** Sara Akram

---

## [Editor Report · Acceptance letter]

PONE-D-25-59942R2

PLOS One

Dear Dr. Trotta,

I'm pleased to inform you that your manuscript has been deemed suitable for publication in PLOS One. Congratulations! Your manuscript is now being handed over to our production team.

Kind regards,

on behalf of

Dr. Muhammad Zammad Aslam

Academic Editor

PLOS One